

# Multi-scale and multi-site resampling of a study area in spatial genetics: implications for flying insect species

Julien M. Haran[1,2], Jean-Pierre Rossi[3], Juan Pajares[4], Luis Bonifacio[5], Pedro Naves[5], Alain Roques[1] and Géraldine Roux[1]

[1] UR633 Zoologie Forestière, INRA, Orléans, France
[2] CBGP, CIRAD, Montpellier SupAgro, INRA, IRD, Univ. Montpellier, Montpellier, France CIRAD, CBGP, Montpellier, France
[3] CBGP, INRA, CIRAD, IRD, Montpellier SupAgro, Univ. Montpellier, INRA, Montpellier, France
[4] Sustainable Forest Management Res Inst, Universidad de Valladolid, Palencia, Spain
[5] Instituto Nacional de Investigacao Agraria e Veterinaria, INIAV, Oeiras, Portugal

Corresponding author
Julien M. Haran,
julien.haran@cirad.fr

## ABSTRACT

The use of multiple sampling areas in landscape genetic analysis has been recognized as a useful way of generalizing the patterns of environmental effects on organism gene flow. It reduces the variability in inference which can be substantially affected by the scale of the study area and its geographic location. However, empirical landscape genetic studies rarely consider multiple sampling areas due to the sampling effort required. In this study, we explored the effects of environmental features on the gene flow of a flying long-horned beetle (*Monochamus galloprovincialis*) using a landscape genetics approach. To account for the unknown scale of gene flow and the multiple local confounding effects of evolutionary history and landscape changes on inference, we developed a way of resampling study areas on multiple scales and in multiple locations (sliding windows) in a single large-scale sampling design. Landscape analyses were conducted in $3*10^4$ study areas ranging in scale from 220 to 1,000 km and spread over 132 locations on the Iberian Peninsula. The resampling approach made it possible to identify the features affecting the gene flow of this species but also showed high variability in inference among the scales and the locations tested, independent of the variation in environmental features. This method provides an opportunity to explore the effects of environmental features on organism gene flow on the whole and reach conclusions about general landscape effects on their dispersal, while limiting the sampling effort to a reasonable level.

## INTRODUCTION

Landscape genetics examines the relationship between landscape and environmental features and genetic structure (*Manel et al., 2003*; *Manel & Holderegger, 2013*). It enables inference about which environmental features facilitate or hinder gene flow (*Zeller, McGarigal & Whiteley, 2012*), which is a key factor for understanding the persistence and evolution of species and populations and has significant consequences for conservation

planning (*Castillo et al., 2014*; *Van Strien et al., 2014*). As an emerging and fast moving field, substantial effort to optimize the method is still required in order to make relevant and optimum inferences (*Anderson et al., 2010*; *Cushman, Shirk & Landguth, 2013*; *Manel & Holderegger, 2013*). Landscape genetic analyses are usually conducted on a single scale and in a single location (*Zeller, McGarigal & Whiteley, 2012*) and therefore provide results that are, strictly speaking, only applicable to the particular area under study (*Short Bull et al., 2011*). In fact, genetic structure is determined by multiple micro- and macro-evolutionary processes acting on different spatial and temporal scales, and is rarely homogeneously distributed across a study species' distribution range (*Waters, Fraser & Hewitt, 2013*). For example, in addition to contemporary or historical environmental effects on dispersal (*Zellmer & Knowles, 2009*), the genetic structure of organisms is often influenced by historic differentiation due to quaternary climate oscillations (*Hewitt, 2000*), or by biased dispersal due to local adaptation to specific environmental conditions (*Sexton, Hangartner & Hoffmann, 2014*; *Pflüger & Balkenhol, 2014*). The diverse factors acting on different temporal and spatial scales may generate genetic patterns that are inconsistent across locations or regions, which results in conflicting signals of environmental factors acting on gene flow. In addition to these multiple limitations, the landscape genetics approach is increasingly being applied to flying organisms with dispersal abilities that are difficult to predict and for which barriers to dispersal may be hard to conceptualize (*Zeller, McGarigal & Whiteley, 2012*). In these situations, an experimental design is difficult to set up a priori, which can lead to limited inference (*Dreier et al., 2014*).

To integrate these factors and deal with the variability of the results in landscape genetic analysis, several authors have highlighted the importance of two aspects: matching the study design to the process being investigated (scale of sampling area, *Cushman & Landguth, 2010*; *Manel & Holderegger, 2013*) and considering landscape level replications (multiple locations of sampling areas, *Short Bull et al., 2011*). The study scale is fundamental in landscape genetics, because species respond to environmental features on a continual range of scales, which may affect correlation coefficients (*Anderson et al., 2010*; *Manel & Holderegger, 2013*). This point has been highlighted in several empirical studies and simulation exercises (*Cushman & Landguth, 2010*; *Angelone, Kienast & Holderegger, 2011*; *Galpern, Manseau & Wilson, 2012*; *Dudaniec et al., 2013*; *Keller, Holderegger & Van Strien, 2013*), in particular for organisms exhibiting wide home-ranges, such as large mammals (*Galpern, Manseau & Wilson, 2012*; *Zeller et al., 2014*). The scale of the study design is also crucial for flying species for which the scale of gene flow may be much larger than expected (*Dreier et al., 2014*). However, landscape genetic studies still rarely consider this aspect (*Zeller, McGarigal & Whiteley, 2012*) and how it affects inference in the detection of general effects of environmental features on dispersal and gene flow. Landscape-level replication is another fundamental aspect in landscape genetics (*Holderegger & Wagner, 2008*). The term replication usually refers to the replication of sampling areas (sampling units; *Short Bull et al., 2011*). Such an experimental design provides a "quantitative" dimension in landscape genetic analysis, allowing conclusions to be drawn about the general effects of landscape features on the dispersal of organisms. These replications are particularly useful to account for multiple local confounding effects of the genetic structure of organisms

detailed above. Few studies have included replication in landscape genetic studies, and the number of replications considered is often small (*Drizen et al., 2007*; *Kindall & Van Manen, 2007*; *Zalewski et al., 2009*; *Short Bull et al., 2011*).

In order to provide a more complete and comprehensive picture of the general effects of the landscape on the dispersal of organisms, there is a need to integrate various scale and landscape level replications of sampling areas in more empirical landscape genetic studies. However, such exploration often remains limited due to the substantial sampling efforts required. Resampling methods (*Sjöstedt-de Luna , 2001*) are an interesting perspective in landscape genetics, as they offer the possibility of examining variation of inference in several sub-parts of a single sampling design, potentially including variations of scale and location of study areas. In our study, we developed a method combining multi-site and a multi-scale resampling of sliding windows (study areas) to explore on which scale and in which locations environmental features fostered or hindered the gene flow of a flying insect species: *Monochamus galloprovincialis* (Coleoptera, Cerambycidae). We first characterized the broad-scale genetic structure of the beetle across the study area, and we used the "isolation-by-resistance" (IBR) framework to model beetle dispersal as a function of three relevant landscape features (temperature, elevation, and pine cover), called IBR hypothesis (IBR-T, IBR-E and IBR-P respectively). For each of these IBR hypothesis, we then carried out a landscape genetic analysis based in 30,576 resampled areas of extents ranging from 220 to 1,000 km and distributed in 132 sampling locations on the Iberian Peninsula. Lastly, we searched for the main landscape features affecting gene flow in *M. galloprovincialis* and we analyzed how the scale and location of the study area impacted inference.

## METHODS

### Sampling and genotyping

The study area covered the entire Iberian Peninsula (582,000 km$^2$) with altitudes ranging from sea level up to 2,444 m. *M. galloprovincialis* specimens were trapped between 2011 and 2013 at 137 sites spread over the Iberian Peninsula. We used multifunnel traps baited with a volatile attractant (Galloprotect, SEDQ, Spain) installed during the summer to catch flying adults. The traps used had a radius of attraction of 100 m (*Jactel et al., 2015*) and were placed in dense pine stands (where beetle density is high; *Jactel et al., 2015*) to limit consanguinity among the individuals caught. After collecting, adults were stored in 96.66% ethanol at 4 °C. Despite intensive trapping, *M. galloprovincialis* was not recorded in the central lowlands of Castilla y Leon, central Galicia and Asturias districts (Fig. S1). We obtained 1,050 individuals at 132 sites, with an average sample per location of 7.68 individuals. Therefore, sampling consisted of a trade-off between the number of localities and the size of demes, keeping the sampling effort at a realistic level. Details of sampling localities and year of collection are given in Table S1. Individuals collected at the same locality were considered as one deme. The distribution of sites covered most of the pines forests found in the Iberian Peninsula (Fig. S1).

DNA was isolated from two legs per individual using a Nucleospin Kit (Macherey-Nagel, Düren, Germany). Specimens were genotyped at 12 microsatellite loci (Mon01,

Mon08, Mon17, Mon23, Mon27, Mon30, Mon31, Mon35, Mon36, Mon41, Mon42 and Mon44; *Haran & Roux-Morabito, 2014*). Details of primer sequences and the protocol for genotyping are given in Table S2 . Results showing negative or ambiguous amplification of particular loci were repeated once and considered null when still unsatisfactory. Individuals exceeding two missing loci were removed for the analysis. Deviation from Hardy Weinberg Equilibrium ($F_{is}$) was estimated for each deme, each inferred cluster and for the whole dataset using GENEPOP 4.2 (*Raymond & Rousset, 1995*). The frequency of null alleles at each locus was tested using FREENA (*Chapuis & Estoup, 2007*) among three large demes ($n > 19$). Loci exceeding a rate of 7% of null alleles across populations were discarded from further analysis. Allelic richness was computed for each deme using rarefaction (HP-RARE; *Kalinowski, 2005*). The absence of linkage disequilibrium between pairs of loci was reported in a previous population-based study (*Haran et al., 2015*).

## Genetic structure

We used the Bayesian approach implemented in STRUCTURE 2.3.4 (*Pritchard, Stephens & Donnelly, 2000*) to identify the main genetic clusters among Iberian demes. STRUCTURE assigns individuals to a predefined number of clusters based on allele composition and linkage disequilibrium. We used the Delta K method (*Evanno, Regnaut & Goudet, 2005*) to determine the number of clusters (K) that best fitted the data. Genotypes were analyzed using default parameters (admixture model, correlated allele frequency). We made ten repeats of a 200,000 burn-in period followed by 500,000 replicates of Markov Chain Monte Carlo (MCMC), for $K$ values ranging from 1 to 20. The results were uploaded in STRUCTURE HARVESTER (*Earl & VonHoldt, 2012*) to determine the optimum K. We also explored the existence of genetic clusters among demes using a principal component analysis (PCA) performed on allele frequencies (Adegenet package; *Jombart, 2008*). To account for potential confounding effects of differentiated genetic clusters (possibly of evolutionary history origin) on the inference of gene flow, the landscape genetic analyses of this study (see below) were performed twice, once within the main cluster identified by STRUCTURE and PCA, and once with the whole dataset including all clusters.

The genotypes of *M. galloprovincialis* were also analyzed taking a spatial approach in order to identify nested levels of genetic structure linked to scales of study. The scores of the sampling locations on axis 1 of the PCA are linear descriptors of the allele frequencies and, as such, can be used as a univariate statistical measure of genetic composition. The scores may encapsulate relevant spatial information, so we explored this point using a specific tool borrowed from geostatistics: the variogram (*Wagner et al., 2005*; *Goovaerts, 1997*). The variogram is used in all branches of life sciences in order to explore spatial patterns and determine the main spatial scales on which structures occur. In our study, we analyzed the score of sample points on axis 1 using a variogram to gain a better understanding of the spatial component of the variation encapsulated in the first axis of the PCA. Let $z(u_\alpha)$, with $\alpha = 1, 2, \ldots, n$, be a set of n values of sample scores on a PCA axis where $u_\alpha$ is the vector of spatial coordinates of the $\alpha$th observation. In geostatistics, spatial dependence is described in terms of dissimilarity between observations expressed as a function of the separating distance (*Goovaerts, 1997*). The average dissimilarity between data separated by a vector h

is measured by the empirical semi-variance $\hat{y}(h)$, which is computed as half of the average squared difference between the data pairs:

$$\hat{\gamma}(h) = \frac{1}{2N(h)} \sum_{x=1}^{N(h)} [z(u_\alpha) - z(u_\alpha h)]^2 \tag{1}$$

where $N(h)$ is the number of data pairs for a given lag vector $h$, $z(u_\alpha)$ and $z(u_\alpha + h)$ the score values of all sample locations separated by a vector $h$. The more alike the observations at points separated by $h$ are, the smaller $\hat{y}(h)$ will be, and vice versa. The plot of $\hat{y}(h)$ against $h$ is called a variogram and represents the average rate of change of $z$ with distance. Its shape describes the pattern of spatial variation in terms of general form, scales and magnitude (*Goovaerts, 1997*).

Variograms are good tools for depicting spatial structures and analyzing nested patterns (*Burrough, 1983*); when structures occur on different spatial scales, the resulting variogram exhibits different plateaus (horizontal flattening of the curve) in association with different scales (*Robertson & Gross, 1994*; *Rossi, 2003*). The range of the variogram is the distance at which the plateau occurs. Multi-plateau variograms exhibit different ranges which provide synthetic information about the spatial scales in play. Readers are referred to *Goovaerts (1997)* for a thorough introduction to variograms and geostatistics and to *Wagner et al. (2005)* and to *Guillot et al. (2009)* for an introduction to this tool in the field of population genetics. Variograms were computed using the R geoR package (*Ribeiro & Diggle, 2001*).

## Landscape genetic analysis

Conventional landscape genetic analyses were first carried out by computing pairwise genetic distances and landscape resistance distances, and then by correlating them.

### Computing pairwise genetic distances

Genetic distances were computed between pairs of individuals using an individual-based metric (*Shirk et al., 2010*; *Prunier et al., 2013*). We first constructed a matrix where each individual was a row and alleles were columns and where genotypes were coded for each allele as 0 when absent, 1 when single at a locus (heterozygotes) or 2 for homozygotes (*Shirk et al., 2010*). Thus, individuals were represented as a linear vector of size n, where n was the total number of alleles encountered in all the individuals genotyped. We then generated a semi matrix of distance between all pairs of individuals. We computed the Bray–Curtis percentage of dissimilarity (*Legendre & Legendre, 1998*) to estimate differentiation between all pairs of individuals. Calculations were performed using the R vegan package (*Oksanen et al., 2016*).

### Computing landscape resistance distances

We then selected the environmental features considered to be the most likely to influence the dispersal of *M. galloprovincialis* given the existing knowledge of species requirements. Apart from Euclidian geographic distances (null model), we considered three environmental features to be potential drivers of dispersal (pine density, temperature, and elevation).

Environmental resistance to dispersal was modeled as a function of pine density as this parameter determines the volume of resource available for *M. galloprovincialis* and is

thought to affect its foraging dispersal. As the dispersal behavior of this beetle in reaction to pine density is not known, we modeled this parameter according to two alternative scenarios. (1) High pine densities are positively correlated with beetle dispersal (scenario hereafter called Pc, "pines as corridors"). In this scenario, a dense pine cover represents a corridor for dispersal due to the large amount of resources available. Conversely, a low pine density would represent a barrier. (2) High pine densities are negatively correlated with beetle dispersal (scenario hereafter called Pr, "pines as resistance"). For this second scenario, it was assumed that a dense pine cover provides sufficient resources for local populations, which would therefore not need to disperse. This scenario assumes increased dispersal in low pine cover areas. To model resistance based on pine density, we considered the sum of densities of all pine species encountered in a grid cell, because in the Iberian Peninsula, *M. galloprovincialis* shows no preference for pine species. As shown in previous studies, this beetle will live in the dead wood of any of the pine species considered in this study (*Pinus pinaster, P. nigra, P. sylvestris, P. halepensis* and *P. radiata*; *Naves, Sousa & Quartau, 2006*; *Haran et al., 2015*; *Haran et al., 2017*). Resistance to dispersal was also modeled as a function of the mean minimum temperature (and its proxy: elevation, hereafter called scenarios T and E respectively), as low summer temperatures tend to inhibit adult flying activity (*Hernández et al., 2011*), and because low winter temperatures are likely to determine the survival or the development rate of *M. galloprovincialis* larval instars (*Naves, Sousa & Quartau, 2006*). We consider that resistance increases when the annual mean minimum temperatures decrease. We kept temperatures and elevation as distinct environmental features for the analysis, as altitude and temperatures are not similarly correlated in the North and South of Spain.

Resistance distances were computed using the gdistance package (*Van Etten, 2012*). Raster layers of environmental features were imported at a resolution of 10 × 10 km. Such a resolution was chosen because the mean flight distance of *M. galloprovincialis* reaches 16 km, based on flight mill experiments (*David et al., 2014*). Temperature data (1950–2000) were downloaded from *Hijmans et al.* (*2005*; http://www.worldclim.org; original resolution: 1 × 1 km), the pine density from *Tröltzsch, Van Brusselen & Schuck* (*2009*; http://www.efi.int/; original resolution: 1 × 1 km) and elevation from ARCGIS 9.3 (ESRI, Redlands, CA, USA; original resolution: 1 × 1 km). Null distances were not encountered at a grain size of 10 × 10 km, as none of the sampling sites fell within neighboring sites in the same grain. As temperature, elevation and pine density are continuous parameters, we did not assign particular resistances to particular values, but directly used the values (except for the scenario Pc for which values were set as negative). Pairwise resistance distances were estimated based on random walk probabilities (*Chandra et al., 1997*; *McRae, 2006*) and computed using the commuteDistance command (gdistance package). Resistance distances were chosen instead of least cost distances (LCD) because they are thought to be more reliable biologically and produce fewer artifacts over long distances (*McRae, 2006*). We constructed a semi matrix of resistance distances between each pair of individuals. Values were normalized to a common scale for further analysis. Collinearity was estimated using the variance inflation factor (VIF) based on the formula $VIF = 1/(1 - R^2)$, where $R^2$ is the r-squared value of regression between variables. VIF values >10 are usually considered

evidence for collinearity between environmental features (*O'Brien, 2007*). We did not detect collinearity between environmental features over the whole area of study (VIF < 1 for all pairwise comparisons).

### *Correlation analysis*

We tested correlations between the response (genetic distances matrix, G) and resistance distances (resistance matrices; Isolation By Resistance: IBR) and the geographic distances (Euclidian geographic distance; Isolation By Distance: IBD) using partial Mantel tests (*Cushman & Landguth, 2010*). Partial Mantel tests measure associations between two distance matrices while partialling out a third distance matrix. We first used simple Mantel tests to correlate IBD with G. We then tested the effect of IBR in partial Mantel tests. Support for the IBR hypothesis was considered when: IBR should be significantly correlated to G after partialling out IBD ($p < 0.05$) and IBD should be non-significant with IBR partialled out ($p \geq 0.05$; *Cushman et al., 2006*). Mantel and partial Mantel tests were performed using the vegan package with $10^3$ permutations. This approach is widely used in the field of landscape genetics (*Cushman et al., 2006*; *Cushman & Landguth, 2010*; *Galpern, Manseau & Wilson, 2012*; *Castillo et al., 2014*) and has been shown to efficiently infer the drivers of gene flow (*Cushman & Landguth, 2010*). However, partial Mantel tests have received criticism regarding their statistical performance (*Guillot & Rousset, 2013*; *Diniz-Filho et al., 2013*), and are therefore preferably used together with complementary approaches such as ordination methods (*Kierepka & Latchi, 2015*). To overcome the potential weakness of partial Mantel tests on our dataset, and to validate the statistical significance of correlations, distance matrices were also regressed using commonality analysis (*Prunier et al., 2014*). This method is based on variance-partitioning and therefore allows the relative importance of the environmental features shaping genetic structure to be estimated, accounting for covariance in the features tested. For the commonality analysis, the response G was regressed onto each resistance matrix separately and each combination using the R yhat package (*Nimon, Oswald & Roberts, 2013*).

## Resampling on multiple scales and at multiple locations

The above Partial Mantels test and commonality analysis were conducted in areas of various spatial scales and in various locations in order to estimate the effects of scale and location on the inference of the landscape genetic analysis for the four IBR hypothesis (IBR-T, IBR-E, IBR-Pr and IBR-Pc). First, nested sampling areas (sliding windows) were generated over the full extent of the Iberian Peninsula. These areas were constructed as circles of diameters ranging from 220 to 1,000 km (steps of 20 km) and centered on each sampling location (scale therefore refers to the extent; *Mayer & Cameron, 2003*). Mantel tests were performed using all individuals found within each defined area. Areas with a diameter below 220 km were not included, because it was too small to gather neighboring demes for Mantel tests in the areas with scattered sampling. Then, the statistical support of the landscape analysis performed in the sampling areas generated were compared against their scale and location. The effect of scale was observed by summing the number of areas with a supported IBR hypothesis and computing their mean Mantel r on each scale. The effect of the geographic

distribution (location) of sampling areas on the detection of the landscape effect on gene flow was examined by mapping areas with a supported IBR hypothesis. The map was obtained by summing the number of times that each individual was included in a sampling area with IBR hypothesis support among all scales. The values obtained were corrected accounting for intrinsic variation due to overlapping sampling areas. Values at each point were interpolated using the Inverse Distance Weighted (IDW) method in ARCGIS 9.3 (ESRI, Redlands, CA, USA). Given that landscape genetic analyses perform better in a contrasting landscape (i.e., high amplitudes of values of resistant features; *Jaquiéry et al., 2011*; *Cushman, Shirk & Landguth, 2013*), we also sought whether or not support for the IBR hypotheses was due to variations in the environmental features tested. The resistance values of raster cells within each sampling area were extracted and the standard deviation (SD) of those values was computed. We then calculated and compared the mean standard deviation of areas with supported and non-supported IBR hypotheses among the area scales tested. Commonality analyses (see above) were performed within each sampling area generated. As for the Mantel tests, the variation in commonality coefficients (percentage of variance explained by a unique and cumulative IBR hypothesis) was observed by changing the scale and location of the sampling areas. The area maximizing commonality coefficients was chosen for representation of the relative importance of environmental features in shaping genetic structure. All computations were performed using R software version 3.0.2 (*R Development Core Team, 2013*).

## RESULTS

### Genotyping

Overall, 1,050 individuals were successfully genotyped. Among the three populations of larger sizes tested ($n > 19$), two loci exhibited substantial null allele frequencies ($>7\%$) and were therefore not considered for further analysis (Mon 01 and Mon 27). A significant heterozygote deficit was detected at four loci (Mon 30, 35, 42, 44). Corresponding null allele frequencies were low ($<7\%$), so these loci were kept. After the removal of incomplete genotypes ($n = 58$) and biased loci, we obtained a total of 992 individuals genotyped at ten loci. The average number of alleles per locus was 10.2 (range: 6–24). The number of alleles per deme (using rarefaction) ranged from 1.32 to 1.64 and $F_{is}$ estimates from $-0.27$ to 0.38 (Table S1).

### Genetic structure

Individuals formed an optimum number of two clusters under STRUCTURE analysis (Fig. S3). The clusters showed a clear geographic structure, exhibiting a split between Portugal and western Galicia (West Iberian cluster) versus the rest of the Iberian Peninsula (East Iberian Cluster; Fig. 1A). The PCA gave similar results on the first axis (eigenvalue: 0.494 accounting for 14.3% of the total inertia), splitting demes into two distinct clusters (Fig. 1C). Estimates of population differentiation ($F_{st}$) between the three populations of large size ($n > 19$) were moderate (Castro Daire /Catsellbell: 0.13; Castro Daire/Vale Feitoso: 0.13; Catsellbell/ Vale Feitoso: 0.05; $p < 0.001$).

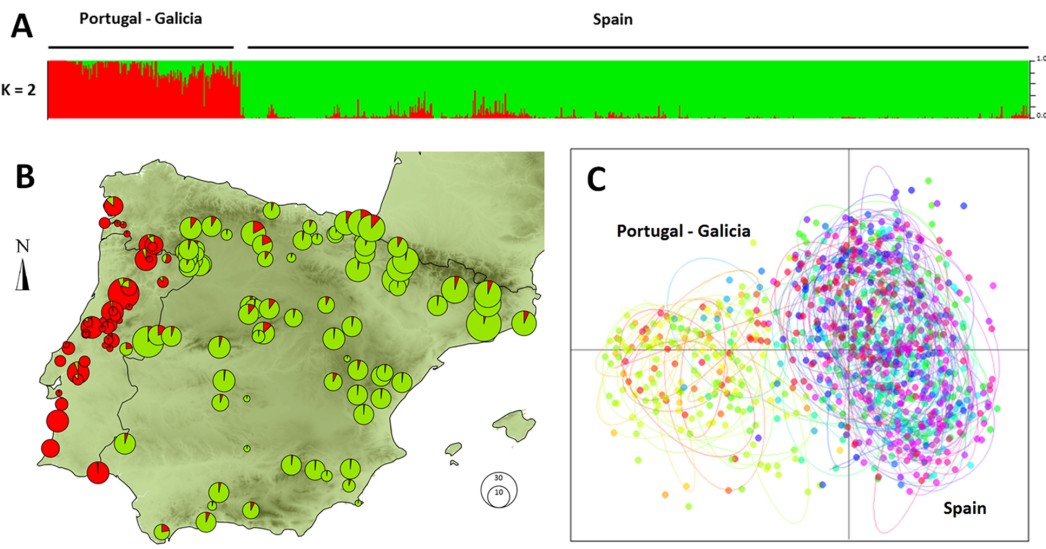

**Figure 1** **Genetic clustering of 992 individuals of *Monochamus galloprovincialis* sampled at 132 locations.** (A) Assignment of individuals to clusters based on a STRUCTURE analysis for $K = 2$. (B) Assignment of demes to clusters for $k = 2$, displayed in geographic context (Iberian Peninsula, the size of the pies refers to the size of the demes). (C) PCA of individuals on first and second axes, colors and ellipses refer to demes.

The data points of the variogram were grouped into 26 distance classes ranging from 0 to 1,252 km, with a distance interval of 50 km. The variogram revealed that the first axis of the PCA corresponded to a highly spatially structured pattern (Fig. 2). The semi-variance first progressively increased with increasing lag distance up to a distance of about 190 km and then reached a plateau. For distances of about 400 km, the semi-variance increased again and leveled off for distances further than 1,000 km. The resulting plateau of the variogram showed the presence of a long-range spatial variation superimposed over a more local, i.e., short-scale genetic structure, occurring on scales of 200 to 400 km. For scales below 200 km, the variogram showed that genotypes were strongly spatially auto-correlated (i.e., non-independent).

## Multiple scale and multiple location analysis

Analyses were conducted on both the whole dataset (992 individuals, 132 localities) and within the East Iberian cluster (790 individuals, 87 localities). Within each dataset, the number of alleles observed across all individuals and loci was 116 and 102, respectively.

Over the whole study area (whole dataset), we generated a total of 30,576 sampling areas. The mean number of individuals within sampling areas varied from 89.18 (SD: 42.42) on the smallest scale (220 km) to 644.58 (SD: 158.07) on the largest scale (1,000 km; Fig. S2). Significant effects of environmental features were detected for all IBR hypotheses tested with partial Mantel tests, but the frequency of areas exhibiting an IBR effect varied between scales and locations (Fig. 3A). Significant effects of environmental features were detected for the IBR-E, IBR-Pr and IBR-T hypotheses in about 15–25% of the areas on the smallest
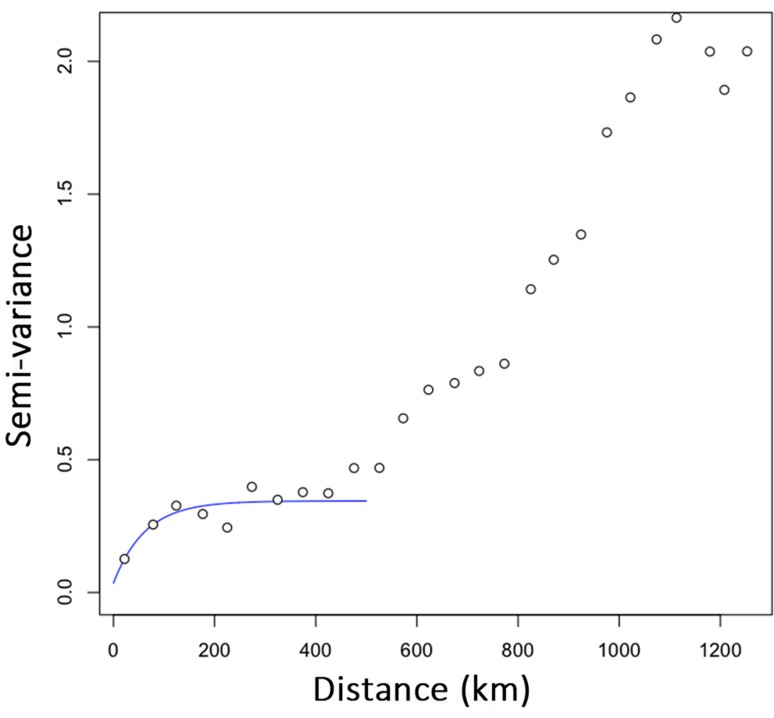

**Figure 2** **Empirical semi-variogram of genotypes of *Monochamus galloprovincialis*.** The variogram was fitted with an exponential model to highlight the first plateau. Data points are shown with a spatial lag distance of 50 km.

scale (220–300 km). The frequency of IBR-E and IBR-Pr gradually increased to reach 97% and 70% for areas of 1,000 km. The frequency of areas with a supported IBR-T hypothesis increased between scales to reach a peak around 600 km (82% of areas) and subsequently decreased. Significant IBR-Pc hypotheses were encountered at a lower frequency. The number of areas with a positive effect of the Pc hypothesis was 4.07 areas, on average, for all the scales considered. Significant isolation by distance (IBD) was observed for 60% of areas on the smallest scale. A first plateau of about 85% of areas was reached for scales ranging between 400 and 700 km, and a second plateau of almost 100% of areas was reached for scales above 700 km. Mean Mantel r values for areas with a supported IBR hypothesis ranked between 0.05 and 0.25 and generally decreased when the scale increased (Fig. 3B). Hypothesis IBR-T showed the highest mean Mantel r value out of all the IBR hypotheses for scales above 360 Km.

Interpolation of supported IBR hypotheses and IBD was based on areas of scales ranging from 220 to 600 km, because most of the variation in the detection of the effects of environmental features was found on these scales (Fig. 3A). For most IBR hypotheses (IBR-E, IBR-Pr and IBR-T) and IBD, the effects were mainly detected in the northern part of the study area (Fig. 4). In contrast, these IBR hypotheses were the least frequently detected on the southern and eastern sides of the Iberian Peninsula. For the IBR-Pc hypothesis, significant effects were detected mainly in Andalucía, along the Betic system. Conversely,

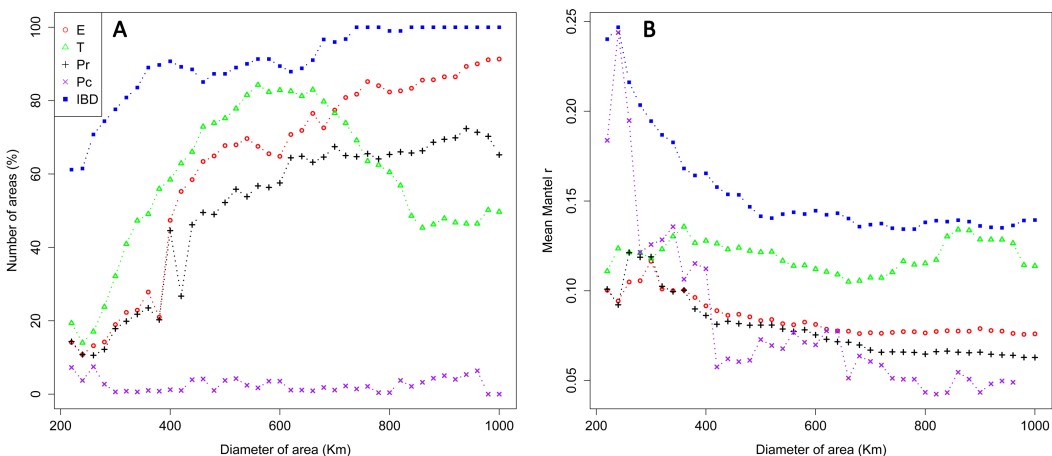

**Figure 3** Percentage of areas with supported IBR hypotheses for Mantel tests (A) and mean partial Mantel *r* (B) of areas with support for the IBR hypotheses ($p < 0.05$) with increasing scales (whole dataset). *E*, Elevation; *T*, Mean minimum temperatures; *Pr* and *Pc*, pine densities as a resistant feature and as a corridor respectively; IBD, Isolation by distance.

low or no effects for this hypothesis were detected in the northern half of the Iberian Peninsula. The distribution of supported hypotheses was similar for those performed on the whole dataset and on the Spanish cluster only (Fig. 4). For hypotheses IBR-E, IBR-Pr and IBR-Pc, the variation of environmental features was lower on average in areas exhibiting significant effects for scales up to 400–600 km (whole dataset; Fig. 5). Above this scale, the mean standard deviation (SD) of areas with a supported IBR hypothesis was either equal to or higher than the mean SD of areas with no support. For the IBR-T hypothesis, the mean SD of areas with support was above the mean for non-supported areas, for most of the scales. The results of the commonality analysis were in agreement with those of the Mantel tests. The IBR-T and IBR-Pr hypotheses purely contributed to most of the total variance explained (20.77 to 32.65% and 21.82 to 35.24%, respectively, Table 1), and the highest contribution to the total variance explained was observed for the joint effects of IBR-E and IBR-T (54.31 to 56.43%). Conversely, the IBR-Pc hypothesis made a limited contribution to the variance explained in pure effect (1.80 to 3.81%). As for the Mantel tests, the sampling areas showing the maximum explained variance over all sampling areas (20.9 to 24%) were located in the western and northern parts of the Iberian Peninsula and had a medium diameter (520 to 620 km).

## DISCUSSION

The landscape genetics approach seeks to infer general drivers of gene flow for species in a heterogeneous landscape context (*Manel & Holderegger, 2013*). However, the ability of this approach to make inferences about the effect of environmental features may vary due to multiple evolutionary processes acting on the genetic structure of organisms on different spatial and temporal scales. In this study, we explored potential barriers against
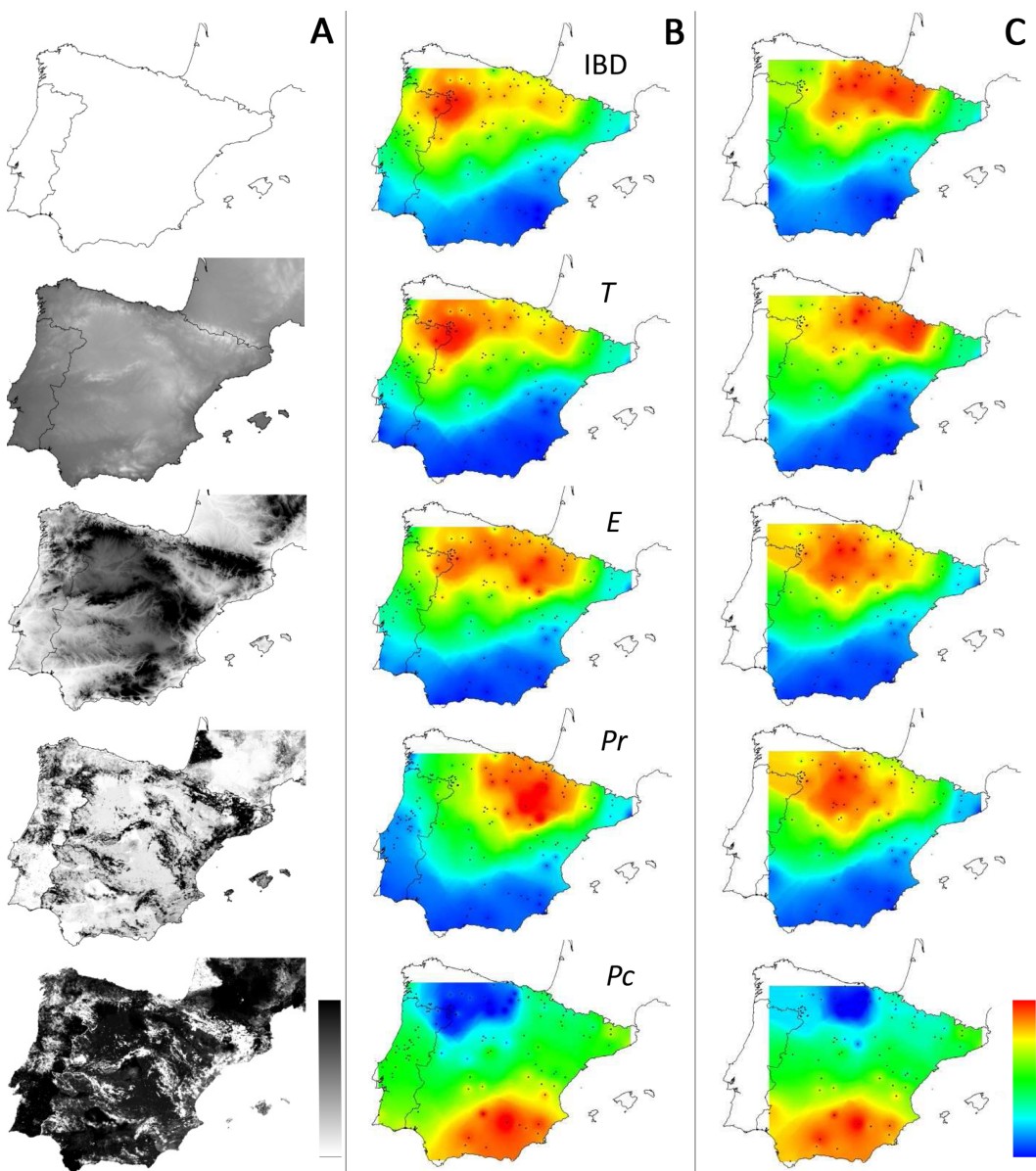

**Figure 4** **Distribution of supported IBR hypotheses through Mantel tests for the environmental features tested.** (IBD: Euclidian distances; *T*, mean minimum temperatures; *E*, elevation; *Pr*, high pine densities as barriers; *Pc*, high pine densities as corridors). Gray maps (A) refer to the distribution of environmental features associated with resistance models (from white to black: low to high resistance values). Colored maps refer to interpolations of supported IBR hypotheses on the whole dataset (B) and within the Eastern Iberian cluster only (C). From blue to red: low to high frequency of study areas with supported resistance models.

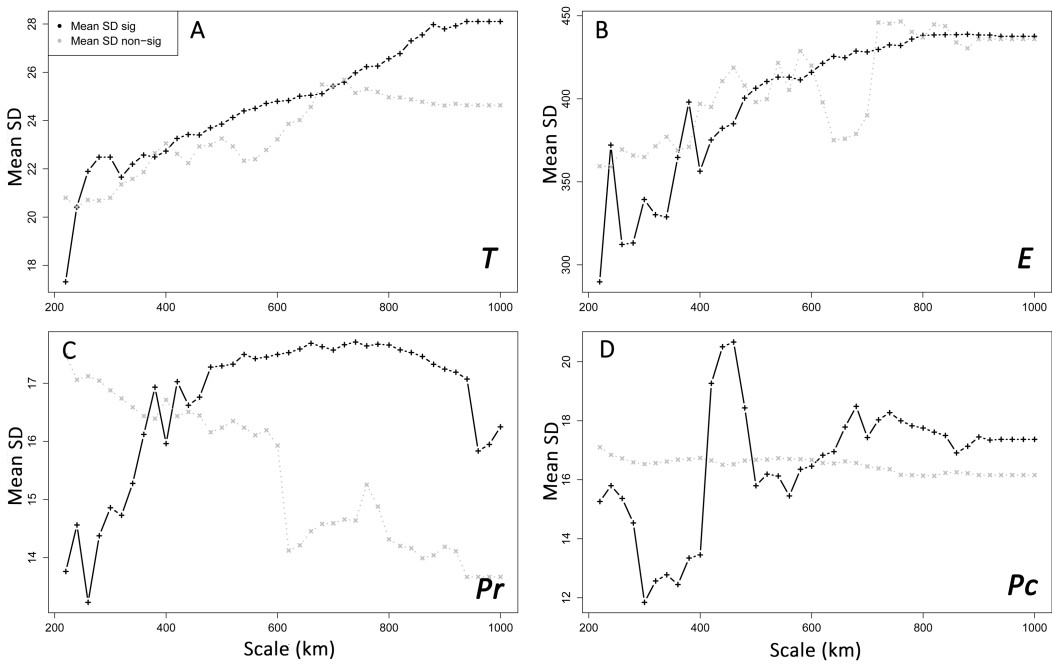

**Figure 5** **Spatial heterogeneity (mean standard deviation, SD) of environmental features in areas with supported and non-supported resistance hypotheses through Mantel tests with increasing scales.** Mean SD: mean standard deviation. *T*, mean minimum temperatures (A); *E*, elevation (B); *Pr*, high pine densities as barriers (C); *Pc*, high pine densities as corridors (D). sig: significant, non-sig: non-significant.

and corridors for the dispersal and gene flow of a flying insect in a large area with dramatic landscape changes. Based on multi-scale and multi-site resampling of study areas, we successfully identified the effects of environmental features on gene flow for independent study areas, and for different inference methods. Resampling provided a quantitative value for the results, making it possible to control the Type-1 errors associated with the inference methods employed (*Short Bull et al., 2011*). Apart from identifying relevant environmental features, the resampling method showed patterns for the frequency at which landscape effects were detected among locations, especially on the smallest spatial scales. Interestingly, most of these patterns could be explained by the biology of the species model, by identified artifacts of the sampling design and by known changes in the landscape structure of the study area.

## Importance of the scale of the study area for flying insects

We observed a notable influence of scale on the detection of supported IBR hypotheses with Mantel tests for most of the environmental features tested (IBR-E, IBR-T and IBR-Pr). Support was scarcely detected on the smallest spatial scale (220–400 km) and was generally more often detected with increasing scale. A larger study area had a larger sampling leading to better detection of positive correlations via partial Mantel tests (*Cushman & Landguth, 2010*; *Landguth et al., 2012*). Beyond the simple effect of the sampling size, the spatial scale of gene flow and resulting genetic structure also influenced the inference

**Table 1 Commonality coefficients of both unique and common effects for the three sampling areas with the highest variance explained.** Code pop: population code for the center of the sampling area. Scale: diameter of sampling area (km). $N$: number of individuals in sampling area. Coef.: percentage of variance explained by environmental features (IBR hypotheses). % Total: percentage of the contribution of environmental features to the total variance explained.

| Code pop | 85 | | 130 | | 131 | |
|---|---|---|---|---|---|---|
| **Scale** | 620 | | 540 | | 520 | |
| **N** | 225 | | 254 | | 244 | |
| **IBR hypotheses** | **Coef.** | **% Total** | **Coef.** | **% Total** | **Coef.** | **% Total** |
| *1st-order* | | | | | | |
| *E* | 0,008 | 3,408 | 0,001 | 0,351 | 0,002 | 0,807 |
| *T* | 0,050 | 20,775 | 0,070 | 32,651 | 0,059 | 28,108 |
| *Pc* | 0,004 | 1,806 | 0,008 | 3,678 | 0,008 | 3,811 |
| *Pr* | 0,085 | 35,235 | 0,047 | 21,817 | 0,046 | 22,214 |
| *2nd-order* | | | | | | |
| *E, T* | 0,136 | 56,426 | 0,117 | 54,314 | 0,115 | 54,953 |
| *E, Pc* | −0,003 | −1,255 | −0,001 | −0,243 | −0,001 | −0,430 |
| *T, Pc* | 0,001 | 0,426 | 0,007 | 3,236 | 0,010 | 4,893 |
| *E, Pr* | −0,002 | −0,897 | 0,031 | 14,375 | 0,020 | 9,721 |
| *T, Pr* | −0,013 | −5,398 | 0,018 | 8,406 | 0,014 | 6,540 |
| *Pc, Pr* | 0,024 | 10,049 | 0,008 | 3,579 | 0,011 | 5,333 |
| *3rd-order* | | | | | | |
| *E,T,Pc* | 0,014 | 5,724 | −0,005 | −2,311 | −0,001 | −0,474 |
| *E, T, Pr* | −0,023 | −9,712 | −0,069 | −32,007 | −0,047 | −22,356 |
| *E, Pc, Pr* | 0,003 | 1,313 | 0,026 | 12,118 | 0,030 | 14,509 |
| *T, Pc, Pr* | −0,009 | −3,795 | 0,006 | 2,961 | 0,010 | 4,950 |
| *E, T, Pc, Pr* | −0,034 | −14,106 | −0,049 | −22,926 | −0,068 | −32,581 |
| Sum | 0,240 | 100 | 0,216 | 100 | 0,209 | 100 |

**Notes.**

*E*, elevation model (high elevations = resistance to dispersal); *Pc*, Pine density model (high pine density = corridors to dispersal; *Pr*, reversed pine density model (high pine density = resistance to dispersal); *T*, temperature model (low minimum annual temperatures = resistance to dispersal).

of the hypothesis tested. The smallest spatial scales of 190–400 km corresponded to the distances at which the variogram showed an initial plateau of genetic dissimilarity (Fig. 2), and at which a low percentage of study areas with a significant effect was observed (Fig. 3). This correspondence suggested that for this range of scales, the dissimilarity between individuals of *M. galloprovincialis* was not enough to show a significant effect of environmental features on gene flow in most study areas. In contrast, the peak (for IBR-T) or inflection of the curves (for IBR-Pr, IBR-E) of the number of areas with supported IBR hypotheses observed on scales ranging from 400 to 600 km corresponded to the increase in dissimilarity in the variogram (Fig. 2). Thus, scales above 400 km seemed more appropriate for observing a genetic structure of *M. galloprovincialis* potentially structured by environmental features. Interestingly, we found that the variation in the frequency of areas with support was specific to each environmental feature tested. In line with previous

studies on large mammals (*Zeller et al., 2014*) and insects (*Rasic & Keyghobadi, 2012*), this study highlighted that each environmental feature affected gene flow on a distinct scale, and that empirical studies including several features should always consider a range of study scales.

Weak support of the IBR hypothesis on a small scale illustrated a general problem of the landscape genetic analysis performed on flying species, which are naturally less affected by environmental features than non-flying species. For such species, the combination of intensive dispersal and associated gene flow and a limited number of environmental features affecting dispersal makes inference difficult on small spatial scales (*Dreier et al., 2014*). In the *Monochamus* species, substantial flying distances have been measured (up to 22 km in the field: *Takasu et al., 2000*; *Linit & Akbulut, 2003*; *Hernández et al., 2011*; *Gallego et al., 2012*; *Mas et al., 2013*; *David et al., 2014*), causing strong inbreeding of populations and leading to a weak local genetic structure (*Kawai et al., 2006*; *Shoda-Kagaya, 2007*; *Haran et al., 2015*). Considering a continuous range of scales in the analysis prevented us from basing our conclusions on a scale for which the effect of environmental features could not be detected. In this respect, an experimental design for flying insect species should follow the approaches used for large mammals, for which multiple scales, including very large scales, have been used to account for a scale of gene flow that is unknown a priori (*Galpern, Manseau & Wilson, 2012*; *Zeller et al., 2014*).

## Effect of geographic location on inference

Based on the resampling of study areas in the East Iberian cluster, we observed the existence of heterogeneous geographic distribution of supported resistance models (IBR-T, IBR-E and IBR-Pr). Most variation in the distribution of support for IBR was observed on small and intermediate scales (220–600 km), the smallest scale being larger than areas at which landscape genetic analysis are usually conducted (*Short Bull et al., 2011*; *Zeller, McGarigal & Whiteley, 2012*). On these scales, supported effects were mainly detected in the northern-central part of the Iberian Peninsula. Conversely, effects were less supported in the rest of the Iberian Peninsula (center, south and coasts). Variation of support are known to occur when multiple area of study are tested in landscape genetics (*Short Bull et al., 2011*), however, in this case all areas with an effect were located in the same region. These results highlighted that the genetic structure of *M. galloprovincialis* was structured according to the environmental features tested in some areas but not in others, independently of variations in the heterogeneity of the environmental features (Fig. 5). This observation is interesting, because one might expect a native species such as *M. galloprovincialis* to have homogeneous dispersal in response to environmental features, at least within a genetic lineage, and that this effect would be detected homogeneously across a study area. Determining the exact origin of such heterogeneity is challenging. It is suggested that this variation was a legacy of changes in the distribution of host trees in the Iberian Peninsula. Indeed, the distribution and density of pine trees have been strongly affected by anthropogenic activities over recent centuries (*Ruiz-Benito, Gomez-Aparicio & Zavala, 2012*; *Lopez-Merino et al., 2014*), resulting in local extinction, as well as the connectivity and fragmentation of pine tree cover

over time. For example, *Abel-Scaad, Lopez-Saez & Pulido (2014)* showed that pine trees locally disappeared from the Central Iberian System during the middle ages. In contrast, these areas were 80% afforested with pines trees over the 1940–1950 period. It is assumed that such recent modifications have dramatically affected the distribution and abundance of *M. galloprovincialis*, and that the time since these modifications occurred is too short to have affected the genetic structure of the beetle according to the environmental features tested (*Epps & Keyghobadi, 2015*).

This study also highlighted empirically how important it is to account for the genetic differentiation of the species derived from evolutionary history when selecting the location of a study area. We found that the effect of environmental features on gene flow was not detected in study areas mainly located along the western Iberian coast, and these areas always overlapped the West and East Iberian genetic clusters. In these cases, substantial genetic differentiation blurred the genetic structure derived from the effect of environmental features on gene flow, which is more recent and weaker in *M. galloprovincialis*. This effect of differentiated genetic clusters on inference was also observed in the strength of the correlations: we observed a decrease in the mean Mantel r for large-scale study areas (600–1,000 km) which always overlapped the two genetic clusters. It is difficult to predict exactly to what extent genetic divergence derived from evolutionary history can contribute, or not, to the detection of an effect of environmental features on gene flow, as this structure can also derive from old and stable barriers to dispersal. While the vast majority of species exhibit differentiated genetic linages derived from evolutionary history (*Hewitt, 2000*), the genetic structure of a species beyond the extent of the study area is rarely explored in landscape genetic analysis (*Zeller, McGarigal & Whiteley, 2012*). Our results suggest that carrying out analyses both within each genetic cluster and over the whole dataset is preferable, to avoid the confounding effect of evolutionary history on landscape genetic analyses.

## Relevance of inference for the model species

This study enabled us to generalize patterns of dispersal for *M. galloprovincialis* and confirmed previous observations made on the biology and ecology of this species. Mantel tests and a commonality analysis supported the hypothesis that elevated areas and their associated colder temperatures constitute barriers to gene flow for *M. galloprovincialis*. This result corroborates the observations made for the congeneric species *M. alternatus* across the Ohu chain of mountains in Japan (*Shoda-Kagaya, 2007*) and on *M. galloprovincialis* across the Pyrenees (*Haran et al., 2015*; *Haran et al., 2017*). In addition, this hypothesis tends to be confirmed by several studies showing that *M. galloprovincialis* larva development and survival (*Naves & De Sousa, 2009*), and its ability to complete its development within one or two years (*Tomminen, 1993*; *Naves et al., 2007*; *Koutroumpa et al., 2008*), are affected by low temperatures. Lastly, the adult flying activity of this beetle was shown to be negatively affected by low daily temperatures (*Hernández et al., 2011*), and to be directed mainly down-hill (*Torres-Vila et al., 2015*). Our analysis also supports the hypothesis that *M. galloprovincialis* exhibits limited dispersal when pine trees are abundant. This species is

known to develop on dead branches stemming from a self-pruning process encountered in pines (*Mäkinen, 1999*). Dead branches represent a resource that is quite well distributed in space and time in pine stands. Such an abundance of resource is thought to cause limited dispersal in adults. The philopatric behavior of *M. galloprovincialis* in relation to available resources is consistent with flight observations of this species in the field (*Torres-Vila et al., 2015*), or with the behavior of the pine processionary moth (*Thaumetopoea pityocampa*), another oligophagous pine–associated insect (*Démolin, 1969*). Conversely, the *Pr* hypothesis suggested that low pine densities are not barriers to dispersal. This is in agreement with the suggestions of *Torres-Vila et al. (2015)* that the dispersal of *M. galloprovincialis* tends to be enhanced across open areas. *Rossi et al. (2016)* showed that areas without pine forests still show a homogeneous distribution of scattered trees planted for ornamental use using observed and simulated data. We suggest that pine trees outside forests provide a scattered but homogeneously distributed resource that allows the dispersal of *M. galloprovincialis* across non-forested areas.

## CONCLUSIONS

Our results show the importance of simultaneously considering a continuous range of scales and multiple locations when exploring the effect of environmental features on dispersal in highly mobile species. Multiple scales allow the effect of environmental features to be inferred at the appropriate extent for each feature tested, while preventing the analysis from being focused on an extent where intensive gene flow makes inference impossible due to the lack of genetic structure. In addition, resampling of the study area across multiple locations helped to identify variation in inference due to conflicting signals in genetic structure, and thereby allowed for generalizing conclusions regarding the effects of environmental features on dispersal and gene flow. As a result, the combination of a resampled study area on multiple spatial scales across various locations in a landscape genetic analysis provides a more general picture of the effects of environmental features on the gene flow of organisms and has the power to reduce the variability of results while limiting the sampling effort.

## ACKNOWLEDGEMENTS

We warmly thank Rolf Holderegger and Bertrand Gauffre for valuable comments on the early versions of this manuscript. We also thank Jérôme Rousselet and Christelle Robinet for interesting exchanges and help regarding the methodology and Peter Biggins for English editing.

# APPENDIX. 1. R SCRIPT DETAILING THE APPROACH USED IN THIS STUDY

```
# Simplified version of the script used in this study. Provides an overview of the general
method employed.
#—————————————————————————————————
# create and plot background matrix with artificial barrier in middle
m <- matrix(1, nrow=10, ncol=10) ; m
m[,5] <- 4

library(raster)
r <- raster(m)
plot(r)

# create and plot transition matrix
library(gdistance)
t <- transition(r, transitionFunction=mean, 4, symm=TRUE, intervalBreaks=3)
plot(raster(t))

# create and plot sampling points and associated genetic data.
# (x coordinates, y coordinates, genetic data for 3 loci)
matG2 <- matrix(c(0.21, 0.22, 0.82, 0.23, 0.81, 0.83, 0.81, 0.21, 0.50, 0.51, 0.23, 0.83, 0,
0, 2, 0, 1, 1, 1, 2, 1, 1,
1, 0, 2, 1, 0, 1, 0, 0), ncol=5)
xcoord<- matG2[, 1] ; ycoord <- matG2[, 2]
P<-cbind(xcoord,ycoord)
points(P)

# construction of moving windows (sampling areas)
library("ade4") ; library("vegan")

# Define the extent of sampling areas and the interval wanted
Min <- 0.7 # Minimum radius of areas wanted
Max <- 0.9 # Maximum radius of areas wanted
Step <- 0.1 # interval wanted

# Loops to test correlations in sampling area on multiple scales and in multiple locations
resultsfinal <- cbind(1,1,1,1,1)
colnames(resultsfinal) <- c("xcoord","Ycoord", "Radius", "MantelR", "Pval")
for(Radius in seq(Min, Max, by = Step)){
    results = NULL
    for(i in 1:length(xcoord)){
        Xcircle <- (xcoord [i] + Radius*cos(seq(0,2*pi,length.out=100)))
        Ycircle <- (ycoord [i] + Radius*sin(seq(0,2*pi,length.out=100)))
        polygon(Xcircle, Ycircle)
```

```
# extract individual data in each sampling are constructed
expr <- point.in.polygon(xcoord,ycoord,Xcircle,Ycircle)
xcoord[expr==1]
ycoord[expr==1]
coordPoly <- cbind (xcoord[expr==1],ycoord[expr==1])

# sort data and compute matrix of basic pairwise euclidian distances (not used
further in this example)
CoordOrder<- coordPoly[order(coordPoly[,1],decreasing=FALSE),]
locOrder<-data.frame(CoordOrder)
DisGeoEucl<-dist(locOrder, method = "euclidean", diag = TRUE, upper = TRUE)

# compute corresponding matrix of genetic distances
listcoord = (1:6)[expr==1]
Genet = NULL

for(h in listcoord){
    tmp <- matG2[(matG2[, 1]==xcoord[h])and(matG2 [, 2]== ycoord[h]), ]
    Genet = rbind(Genet,tmp)
        }
GenetOrder<- Genet[order(Genet[,1],decreasing=FALSE),]
GenetOrderSanscoord <- GenetOrder[,-c(1,2)]
MatdistGenet<- vegdist(GenetOrderSanscoord, method="bray", binary=FALSE,
diag=FALSE, upper=TRUE, na.rm = TRUE)
MatdistGenet <- as.dist(MatdistGenet)

# Compute matrix landscape "resistance" distances based on raster
spatiallocX <- locOrder[,1]
spatiallocY <- locOrder[,2]
SpaLoc <- SpatialPoints(cbind(spatiallocX, spatiallocY))
Resdis<- commuteDistance(t, SpaLoc)
Resdis<-as.dist(Resdis, diag = TRUE, upper=TRUE)

# simple mantels test between genetic and landscape "resistance" distances
MantelpRes <- mantel.rtest(MatdistGenet, Resdis, nrepet = 99)
results <- rbind (results, cbind (xcoord [i], ycoord [i],Radius, MantelpRes[2],
MantelpRes[4]))
    }
    resultsfinal <- rbind(resultsfinal,results)
}

# display result file with for each individual: x and y coordinates, radius of sampling
area, mantel output and associated p-value
Resultsfinal
```
### Funding
This work was supported by the European project REPHRAME KBBE.2010.1.4-09 (FP7 Project, Analysis of the potential of the pine wood nematode (*Bursaphelenchus xylophilus*) to spread, survive and cause pine wilt in European coniferous forests in support of EU plant health policy). The first author was funded by the French Ministry of Research and Education. Field work was supported by COST Action FP1002 (COST-STSM-FP1002-14177). The funders had no role in study design, data collection and analysis, decision to publish, or preparation of the manuscript.

### Grant Disclosures
The following grant information was disclosed by the authors:
European project REPHRAME: KBBE.2010.1.4-09.
French Ministry of Research and Education.
COST Action FP1002: COST-STSM-FP1002-14177.

### Competing Interests
The authors declare there are no competing interests.

### Author Contributions
- Julien M. Haran conceived and designed the experiments, performed the experiments, analyzed the data, wrote the paper, prepared figures and/or tables, reviewed drafts of the paper.
- Jean-Pierre Rossi performed the experiments, analyzed the data, wrote the paper, prepared figures and/or tables, reviewed drafts of the paper.
- Juan Pajares, Luis Bonifacio and Pedro Naves conceived and designed the experiments, reviewed drafts of the paper.
- Alain Roques wrote the paper, reviewed drafts of the paper.
- Géraldine Roux analyzed the data, wrote the paper, reviewed drafts of the paper.

### Data Availability
  The R script, sampling locations and microsatellite genotypes have been uploaded as Supplemental Files.

### Supplemental Information
Supplemental information for this article can be found online at http://dx.doi.org/10.7717/peerj.4135#supplemental-information.

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
