# Peer review of "Multi-scale and multi-site resampling of a study area in spatial genetics: implications for flying insect species"

_PeerJ, doi:10.7717/peerj.4135_

## Round 0.1 · original submission · Major Revisions

You are fortunate to have detailed comments from three highly qualified reviewers. They raise a number of general, as well as specific, issues that you must address address in your revision. Please prepare a detailed summary of how you have addressed the reviewers' comments and submit it with your revisions.

In addition, it is essential that you find a colleague who is a fluent, native English speaker and work with her/him to be certain that your text is clear and understandable. Two of the reviewers specifically mentioned this problem, which was conspicuous to me as I read the manuscript. It is critical that readers be able to understand your text.

·

Basic reporting

This manuscript had some grammatical errors, some jargon not defined, and colloquial English used in places (see General comments to author). It would benefit from being read by a native English speaker. I highlighted areas where additional references would be helpful in the General comments to author section.

Experimental design

In general this study seems well-executed, but research questions need to be more specifically defined in the introduction. In the methods section the analyses that pertain to those questions need to be linked to those questions.

Validity of the findings

Results need to be clarified at times (see general comments to author).

Additional comments

This is an interesting study that investigates how varying the scale over which landscape genetic analysis is performed influences results. I do worry that the authors’ reliance on Mantel tests muddies the inferences that can be drawn from the results. In addition to acknowledging the limitations of Mantel tests, and performing other analyses to validate their results, the authors should discuss these limitations and the possible effects on their results in more detail in the discussion section. They do discuss how limitations of Mantel tests can lead to erroneous inferences in a specific case on line 498, but they should address caveats of Mantel tests more generally with regards to their results.

At times I found the writing confusing. In addition to having a native English speaker read the manuscript, the authors need to take care to present the actual results and not make general statements like those on lines 356: “Best values were observed” or 374: “best values were obtained” (see additional examples below).

One place that clarity must be improved is in the analyses described in the section “Multiple scales and multiple locations analysis”, because this seems to be the crux of this study. At first read I was left wondering exactly what the authors intended by these analyses. The authors need to clearly state which analyses pertain to which question, and what information would be gleaned from various potential results. The link between this paragraph and the results presented in Figure 3 was not immediately clear. The authors need to explicitly state what association they investigated, instead of using vague terms like “we tracked the evolution of” (line 280). The analysis presented in the second panel of Figure 3 is more straightforward, but it would be nice for the authors to set this analysis up in the introduction by telling us why it is important to investigate if correlation coefficients change over different geographic scales.

In addition, the authors need to take care to put their results in the context of others’ work in the discussion (especially in the discussion of how Mantel correlation statistics change with scale).

Specific comments are below:

Introduction:

I found the introduction interesting, and I think the authors could highlight the strengths of this study even more strongly by organizing them into their own paragraphs (for example one paragraph that explicitly focuses on scale and one that focuses on the value of replication across a large study area). These points come across as it is currently written but I think this would clarify the strengths of this study even more strongly.

Line 58: The landscape genetic toolbox has greatly expanded in recent years, I’d remove “the landscape genetic toolbox is far from being established” from this sentence.

Lines 93 - 95: needs citation as currently stated

Line 100: remove “As a study system”

Line 102: Write out the number

Line 104: Move to methods

Last paragraph: Instead of a description of the species (most of which could be moved to the methods section), it would be helpful for the authors to outline a few questions they focused on (for readers that are unfamiliar with LG analyses).

Line 124: If the authors give specifics of localities they should refer to a map or table of locations.

Line 124: remove “a sampling of”

Line 125: give units. Also, the authors need to describe why 10 is relevant.

Lines 127 & 129: do the authors need to refer both to figure/table and write “supporting information”? The names make it clear this is supporting information.

Line 132: what publication are these loci described in

Line 142: remove “already”

Line 151: should be “allele”

Lines 181-183: It would be helpful for the authors to explain what is a plateau.

Line 815: It might be useful for the authors to also refer readers to Guillot et al. (2009, Molecular Ecology) here.

Line 201 should read “temperature”

Line 204: There is a typo.

Lines 212-215: I found this confusing. It would be more clear if the authors said something like “M. galloprovincialis shows no preference for pine species. They will live in the dead wood of any species, which tend to include XXX on the Iberian peninsula”.

Lines 216-221: This is repetitive, please make more succinct.

Line 223 needs a citation

Line 232 should read “mill”

Line 244: Did the authors also test for colinearity when smaller areas were considered?

Lines 272-299: It would be helpful for the authors to introduce this section by describing the question they are addressing, and then to describe what insight various potential patterns would provide. They say on line 280 that they tracked the evolution of areas. What does this mean? In the results they present data on how the number of areas with a significant Mantel result changes with size of area investigated. How does this address the question originally posed in the introduction of does genetic structure or the influence of landscape features vary spatially?

Line 276: Please clarify what the sentence beginning “scale dimension” means.

Line 277: replace “between” with “using”

Line 280: The authors need to describe what unbalancing and a supported hypothesis mean.

Line 281-285: I don’t understand how this gives the geographic distribution of areas, please clarify.

Line 295: Do the authors mean “cumulative”?

Line 313: If the authors are going to provide the delta k values they should explain what they mean in the methods section (I don’t think they are necessary).

Line 321: The authors should describe to which analysis this refers

It seems lines 326-329 belong in the discussion. Also, it would be helpful for the authors to describe what these and other variogram patterns would mean in the methods section.

Line 333: what does “for a total of 116 and 102 alleles” mean/refer to?

Line 334: Can the authors provide some evidence of this? This could also be moved to the methods section, and results presented only for 10x10.

Line 343: The authors should move general statements like this to the discussion and focus on specific results here.

The sub-headings throughout the manuscript are helpful but it would be nice for them to match between sections (where is Multiple scales and multiple locations analysis in the results?)

Line 346: Give the actual percentages

Line 350: The authors need to clarify what this sentence means and to which analysis this refers.

Line 356: What is the “best” r? Please re-word. Also, there is a typo here.

Paragraph starting on line 358: If the authors refer to specific places they should show them on a map.

Line 368: The authors should keep their terms consistent, they use significant area, then non-supported area.

Line 380: Why “common”?

Line 383: This summary paragraph is helpful but the first sentence needs to be re-worded for clarity and cited.

Line 387: The comment about environmental features appearing at different times can be removed.

Line 391: The authors should use the words large and small (not low)

Line 397: cite Figure 2 at the end of this sentence

Line 403: “To gather a genetic structure” doesn’t make sense, please re-word.

Lines 394-466 seem unorganized and should be re-structured to clearly discuss each point laid out in the results section. It would be helpful for the authors to add topic sentences instead of just continuously laying out points. For example, the paragraph beginning on line 408 should begin with a sentence about what it is about.

Line 425: Please remove “deal with” and re-word

Line 428: This is a clear topic sentence

Lines 469-478: How does this relate to what others have found?

Typo on line 495 (capitalization)

Table 1: Abbreviations need to be defined in captions of other tables/figures (please take care to do that), so I do not think this table is necessary.

Figure 2: The authors need to describe a semi-variogram in the methods section.

Figure 3: The caption needs to clearly state what variable is on the y-axis. “Development” is confusing, please re-state this.

Figure 5: The authors should change “sign” to “sig”, and define the abbreviations.

Table S1: Where is the description of what the abbreviations mean?

·

Basic reporting

## Clarity of writing, English
In general the writing is clear. However, the English could be improved -- there are some dubious word choices and grammatical mistakes which hamper readability. Here are some examples:

Line 38: what does "to track the variation of inference" mean?

Line 42: should be "Detection of an effect of environmental features on gene flow"

Line 54: "It allows which ... to be inferred" should be "It allows inference about which ..."

Line 94: should be "...picture of the general effects of landscape on the dispersal..."

Line 98 (also 532): don't you mean "variability" instead of "versatility"? Or do you mean "has the power to *increase* versatility"

Line 402-404: what does "gather a genetic structure" mean? In general I don't understand this sentence.

Line 405 (and other places): what does "development of the variation of frequency of areas" mean?

Line 419: "important" doesn't seem like the correct adjective

Also check for incorrect and missing pluralization throughout.

## Context, references
The landscape genetic analyses used in the manuscript are typical of the norms for the field, and the references for the methods (as well as general analytical concerns specific to the field) are sufficient. Study-system-specific references and context are also sufficient. However, the method proposed in the paper is (at least superficially) similar to local regression methods and geographically weighted regression, both of which have been used in multiscale analysis (although not with pairwise data) -- in my opinion more attention could be paid to relevant statistical references outside the field of landscape genetics. A place to start looking might be:
Brunsdon, Fotheringham, Charlton. 1996. Geographically weighted regression: a method for exploring spatial nonstationarity. Geographical Analysis

## Figures, tables
The figures could be made more aesthetically pleasing by increasing resolution and reducing white space, but are generally clear ... although I suggest the following minor improvements for readability:
Figure 2: decrease size of plot relative to axis titles, capitalize axis titles
Figure 4: adding boxes around each panel would help with visual navigation. Also, a color scale would be informative -- what exactly do "low frequency" and "high frequency" mean?
Figure 5: units on x-axis, shorten titles substantially

## Structure, data
The general structure follows PeerJ standards, is self-contained, and the data (georeferenced microsat genotypes) is included in the supplement (as is the R code used to generate the results in the paper).

Experimental design

## Relevance, originality of research question
The authors correctly identify that while environmental effects on animal movement are often scale dependent, yet few studies explicitly (or quantitatively) consider scale. The method presented here is novel to the field, to the best of my knowledge, and in my opinion contributes a useful idea about how to identify and describe scale dependence in isolation-by-resistance analyses.

## Rigor
The insect sampling is extensive, and the field and molecular methods are sufficiently detailed and rigorous.

## Description of methods
The landscape genetic analyses are sufficiently detailed; software and specific functions are referenced throughout. However, the goals of these analyses (and how they fit into the paper at large) were not clear until I read the results. I suggest that the introduction should contain the system-specific questions and hypotheses, and should lay a roadmap for the subsequent descriptions of analyses. For example, adding something to the effect of: "We first characterize the broad-scale genetic structure of the beetle across the study area. We then used the "isolation-by-resistance" (IBR) framework to model beetle dispersal as a function of temperature, elevation, and pine density. By subsetting the data to particular locations and spatial scales and repeating the analysis, we identify how the influence of environment on spatial genetic structure varies with spatial location and scale."

On this note, the last paragraph of the introduction belongs in the methods. Including relevant hypotheses and the motivating questions in the beetle system is fine in the introduction, but detailed description of the study system should be moved to the methods.

My main concern with the paper is that the methodological goal is not clear. There seem to be two intermixed goals. First, to determine what constitutes a sufficient sampling design (e.g. spatial scale) to detect an effect of environment on dispersal; in other words, what scale is sufficient to use in a hypothetical study to infer an effect. Second, to develop a method for investigating and visualizing scale dependence in the effect of environment on dispersal; in other words, given a single large-scale dataset, to determine at what scales an effect is present. While these goals are superficially related, they require different presentations, and in my opinion, different statistical support (details below). It should be clear from the outset which goal (one, the other, or both) this paper is addressing.

The first goal (study design re: spatial scale) is suggested by passages like the following (not a comprehensive list):

Lines 42-43 (Abstract): "Detection of environmental features on gene flow generally increased with an increasing scale of study"

Lines 394-400 (Discussion): "We observed a notable influence of scale on the detection of supported IBR hypotheses ... This correspondence suggested that at this range of scales, dissimiliarity between individuals was often not appropriate to show a significant effect of environmental features on gene flow"

Lines 476-478: "our results show a tradeoff between the sampling of small areas where effects of environmental feastures are strong but scarcely detected and the sampling of large surfaces, where this effect is weaker but often detected"

In my opinion, statements regarding optimal design with regards to "detection" of effects need to be backed up by a simulation study. There are three main reasons for this: (1) it is impossible to say if an effect is detected unless one knowns a priori whether it is true or not (clearly only possible with simulation). (2) With the current setup, sample size is confounded with spatial scale: smaller scales will contain fewer individuals, and a trivial consequence is that inference will be more "noisy" and less powerful. In a simulation study, one can keep sample size constant across spatial scales. (3) On any given dataset, it's difficult to say to what degree the results are artifacts of the particular spatial distribution of samples. I'm not saying that the real-world dataset used in the paper is useless for this goal (it's an excellent worked example), but I think the addition of a simulation study is necessary if the authors want to make statements about "detection" of effects with regards to study design.

However, if the paper is simply trying to illustrate an idea for a new exploratory statistical method, then I don't think simulation is necessary (although it would certainly add credibility).

On this note, the goal of the method as stated in the abstract isn't very clear: "a method for resampling of study areas at multiple scales and multiple locations (sliding windows) to track the variation of inference in spatial genetics". Based on the rest of the paper, it seems like the method is designed to determine what scales/locations exhibit a significant effect of the environment on spatial genetic structure, and I'd suggest rewording along these lines. In general, the methodological goal should be made much clearer in the abstract.

Validity of the findings

Appropriate care seems to have been taken with regards to assumptions underlying the molecular data (linkage disequillibrium, HWE, null alleles, etc.) The authors have been careful to employ multiple methods and address already published concerns about typical landscape genetic analyses.

One concern I have with regard to the proposed resampling method: no mention is made of controlling for Type-I errors, and how these might interact with sample size to influence the results. The goal is to determine if there's an effect of environment on dispersal across a set of locations and spatial scales. Effectively, this is accomplished by subsetting the data, but because the subsets of data across locations/scales overlap, null hypothesis tests across these will not be independent. If trying to aggregate these results over space to get a sense of what effects are most important at what scales, regardless of geography (as is done in figure 3A), then inflation of Type-I error will bias the percent of areas with significant isolation-by-resistance upward. The result could be very misleading if Type-I error varies systematically with spatial scale (for example, because tests at larger spatial scales share more of the same sampling locations).

Also, how might edge effects bias this method at locations along the margin of the study area?

In general, interpolating effect size (ie. Mantel r) onto maps might be more informative than interpolating frequency of rejected null hypotheses. One way to show both is to use color for interpolated effect size, and contour lines to show regions wherein p<0.05.

Finally, I'd suggest that the authors restructure the discussion: currently, it reads like two different papers (with the method-specific discussion in the first half, and the system-specific discussion in the second half). It would be more clear and more effective, in my opinion, to integrate the two sections; so as to show how the proposed method gives biological insight into the long-horned beetle system that would not obtained otherwise.

Reviewer 3 ·

Basic reporting

This manuscript reports the genetic structure of Momochamus galloprovincialis in the Iberian Peninsula. The authors detected that the effects of environmental features differed among spatial scales, suggesting that the combination of a resampled study area at multiple spatial scales across various locations provide a more general picture of the effects of environmental feature.
I highly evaluate the authors’ motivation for contributing the landscape genetics study, but dataset of M. galloprovincialis seems difficult to be discussed in context of this study. Kawai et al. (2006) showed that new mutualism system between Monochamus species and the pine wood nematode destroy original genetic structure formed by historical and environmental effects. I don’t know exact damaged areas of pine wilt disease in the Iberian Peninsula, but invasion of the nematode should affect the genetic structure of the beetle, i.e. they should not be in genetic equilibrium in the damaged area. Artificial movement of the beetle due to damage protection is also problematic to study the genetic structure. In such a situation, discussion of the land scape genetics must be done in more careful manner, at least divide the dames from damaged from undamaged areas.

Experimental design

Collecting the beetles throughout the Iberian Peninsula meets the aim of this study to show the gene flow there.
Many of the individual numbers from one dame were not enough for population genetics analyses, but resampling method could make up the results. This method sounds adequate.

Validity of the findings

Authors adopted well considered and sophisticated statistical methods. The results are well stated if the effect of invasion the nematode can be set aside.

Additional comments

-I think the main aim of this study is measuring gene flow of the beetle, not presenting a method for resampling of study areas at multiple scales and multiple location.
-Figure captions are not kind to readers. For example, what do you indicate by colors and ellipse of Figure1C? They should be more concise.
- “el al.” must not be in italics.

---

## Round 0.2 · Minor Revisions

Your original reviewers have generously provided additional comments, and feel that the revision is a major improvement over the original submission. Two of the reviewers have provided specific comments and suggestions that you will need to address in your revision. Please also have several colleagues carefully proofread the next (and hopefully final) revision before you resubmit it.

·

Basic reporting

Lines 35 & 48 should read "gene flow"

The introduction, methods and results are much clearer. A few comments on the discussion:

Line 410: This is confusing, please re-word.

Lines 420 & 450: Please state this as cause and effect. "A genetic structure" doesn't make sense, and line 450 is repetitive.

Line 436: "geographic scales" would be better than "study scales"

Line 456: "causes" would be better than "origin"

Line 468: This is confusing, please re-word.

Line 495: affected how?

Line 510: "We speculate that" or "It is possible that" would be better than "we suggest".

Experimental design

no comment

Validity of the findings

no comment

Additional comments

This manuscript has greatly improved.

·

Basic reporting

The English is improved from the first submission, and the meaning is clear although there are still a few confusing word choices: I have some minor suggestions on word choice/grammar, listed line-by-line under "comments to authors".

Experimental design

The authors have substantially clarified the goals of this study. I have a few minor comments/questions about methodology that are listed line-by-line under "comments to authors".

Validity of the findings

No additional comments ... I am satisfied by the clarifying rewrites to the discussion and the authors' responses to my questions

Additional comments

Some minor comments/questions/suggestions, line-by-line:

65: What is meant by "method"? "Discipline" or "framework" seems more appropriate

66: "optimal" not "optimum". And what exactly is being optimized?

82: "limited" in what sense? Do you mean "incorrect inferences"?

109-110: Clarify ... "_spatial_ resampling methods". Also a statistical method is not a "perspective", per se. How about something like: "Spatial resampling methods (...) provide a technique to examine variation in inferences across geographic subsets of a single sampling design ..."

160: Language is a little atypical, instead consider something like: "To check for convergence of the Markov chain Monte Carlo algorithm implemented by STRUCTURE, we ran ten independent Markov chains for 500,000 iterations each after an initial burn-in period of 200,000 iterations"

251: I'm confused by "values were set as negative". Do you mean reciprocal instead of negative? Assuming that edge weights (resistances) are calculated by averaging values of adjacent cells, then negative cell values would mean negative transition rates in the random walk, which doesn't make sense.

285: "Mantel" not "Mantels"

285: "areas of various spatial scales" doesn't make sense here; how about: "conducted across various spatial scales and locations"

293 (and elsewhere): "landscape analysis" is vague. Specifically, you are referring to the tests of IBD/IBR?

294: "generated sampling areas" not "sampling areas generated"

319: delete "tested"

328: Sentence is a bit garbled, use something like: "In the STRUCTURE analysis, the optimal number of admixed clusters was two (Figure S3)"

334: What's the test here? ANOVA of the first PCA axis as a function of cluster identity?

359: By positive effect, do you mean positive effect size (like a positive Mantel r)? Or do you mean a significant effect? The wording doesn't make sense as a hypothesis can't have an effect

361: "plateau" of what? The meaning of this sentence isn't clear

372: Does "low effects" mean a low effect size, or a low frequency of significant IBR tests?

373: "_spatial_ distribution of supported hypotheses"

391: Remove "context"

395: How about "dramatic variation in landscape features" instead of "dramatic landscape changes"

396-397: This sentence is vague ... What does "successfully" mean in this context? Is "success" finding significant IBD/IBR? I think what you mean is something like "across different inference methods, we successfully detected variation in the effects of environmental features on gene flow across spatial locations."

402: Change "could be explained by" to "are consistent with"

403: "by _known_ artifacts of the sampling design"

403-404: What does "known changes in the landscape structure" mean? Do you mean systematic spatial variation in the landscape variables?

407-408: "Detection of supported hypotheses" sounds strange and is unclear. I think you mean that patterns that were evident at large spatial scales/sample sizes (and so presumably real) were not found at small scales/sample sizes. You can reword to make this clearer. For example, wording like "Support was scarcely detected on the smallest spatial scale" could be changed to "Hypotheses were rarely supported at the smallest spatial scale".

410: "larger sampling" should be "larger sample size"

413: "the inference of the hypothesis tested" should be "inference about the tested hypotheses"

448-449: Check grammar here

450: "an effect" is too vague in this context. Effect of what?

457: "It is suggested" should be "We suggest"

460: "as well as _influencing_ the _continuity_ and fragmentation of pine tree cover ..."

468: "genetic differentiation of the species _due_ to evolutionary history"

479: as for line 468

505: Do you mean "_the statistical support for_ the Pr hypothesis _in this study_ suggests that ..."

Reviewer 3 ·

Basic reporting

I found this resubmitted manuscript by Haran et al. to be well improved. I recommend it to be accepted.

Experimental design

This study seems well executed.

Validity of the findings

This study sounds.

---

## Round 0.3 · Minor Revisions

The methods and results are much clearer and understandable in this revision. However, the overall conceptual framework for the analysis needs to be strengthened. Specifically, it should be introduced as explicit hypotheses much earlier in the manuscript, in the range of lines 121-127.

These hypotheses obviously include Isolation by Distance (thereafter abbreviated as IBD) as a null hypothesis in relation to Isolation by Resistance (thereafter abbreviated as IBR). The three hypothesized IBR resistance mechanisms - Temperature (T), Elevation (E) which is primarily a proxy for temperature, and Pine Density (Pr as a resistance effect) plus Pc as a positive effect of Pine Corridors. The authors should consider whether it makes more sense to refer to E,T,Pr, and Pc as "hypotheses" or as specific mechanisms for the operation of Isolation by Resistance. In either case, they should be clearly and explicitly stated prior to the Methods section.

The abbreviations E, Pr, and T first appear (in italics) with no prior definition or description in line 377 (67% of the way through the manuscript) without ever having been described or identified as the specific mechanisms for which they are abbreviations. Subsequently these abbreviations appear (either singly or as subsets of (T,E,Pr,Pc) at least 14 additional times in the last third of the Ms. These mechanisms are clearly a major component of this study, and should not suddenly appear in abbreviated form 2/3 of the way through the manuscript.

I suggest using the convention IBR-T, IBR-E, and IBR-Pr when referring to these hypothesized mechanisms. IBR-Pc may require more explanation. It makes no difference whether the notations T,E, Pr, and Pc are presented in italics or in plain text, as long as the usage is consistent.

---

## Round 0.4 · accepted · Accept

I am very pleased with the way this manuscript has turned out. I appreciate the patience and diligence of the authors in thoroughly addressing the suggestions of the reviewers, as well as my own suggestions for improving conceptual clarity. I think that ecologists interested in spatial dynamics and gene flow will find this research very useful.

---

## Author Rebuttal · Round 0.4

**Dear Editor,**

**Indeed, IBR hypothesis were not presented correctly in the last version of the MS. Thanks for this suggestion. In the new version, IBR hypotheses are introduced at the end of the introduction with the other main points of the methodology. All mentions of "hypothesis" were replaced as IBR-T, IBR-E… and italic was removed. The meaning of abbreviations was detailed in the method section.**

**Regards,**

**Julien Haran**

########################

The methods and results are much clearer and understandable in this revision. However, the overall conceptual framework for the analysis needs to be strengthened. Specifically, it should be introduced as explicit hypotheses much earlier in the manuscript, in the range of lines 121-127.

These hypotheses obviously include Isolation by Distance (thereafter abbreviated as IBD) as a null hypothesis in relation to Isolation by Resistance (thereafter abbreviated as IBR). The three hypothesized IBR resistance mechanisms - Temperature (T), Elevation (E) which is primarily a proxy for temperature, and Pine Density (Pr as a resistance effect) plus Pc as a positive effect of Pine Corridors. The authors should consider whether it makes more sense to refer to E,T,Pr, and Pc as "hypotheses" or as specific mechanisms for the operation of Isolation by Resistance. In either case, they should be clearly and explicitly stated prior to the Methods section.

The abbreviations E, Pr, and T first appear (in italics) with no prior definition or description in line 377 (67% of the way through the manuscript) without ever having been described or identified as the specific mechanisms for which they are abbreviations. Subsequently these abbreviations appear (either singly or as subsets of (T,E,Pr,Pc) at least 14 additional times in the last third of the Ms. These mechanisms are clearly a major component of this study, and should not suddenly appear in abbreviated form 2/3 of the way through the manuscript.

I suggest using the convention IBR-T, IBR-E, and IBR-Pr when referring to these hypothesized mechanisms. IBR-Pc may require more explanation. It makes no difference whether the notations T,E, Pr, and Pc are presented in italics or in plain text, as long as the usage is consistent.